# Nonlinear Neutral Delay Differential Equations of Fourth-Order: Oscillation of Solutions

**DOI:** 10.3390/e23020129

**Published:** 2021-01-20

**Authors:** Ravi P. Agarwal, Omar Bazighifan, Maria Alessandra Ragusa

**Affiliations:** 1Department of Mathematics, Texas A and M University, Kingsville, TX 78363, USA; Ravi.Agarwal@tamuk.edu; 2Department of Mathematics, Faculty of Science, Hadhramout University, Hadhramout 50512, Yemen; o.bazighifan@gmail.com; 3Department of Mathematics, Faculty of Education, Seiyun University, Hadhramout 50512, Yemen; 4Department of Mathematics and Computer Science, University of Catania, 95124 Catania, Italy; 5RUDN University, 6 Miklukho, Maklay St, Moscow 117198, Russia

**Keywords:** neutral differential equations, oscillation, fourth-order differential equation, p-Laplacian equations

## Abstract

The objective of this paper is to study oscillation of fourth-order neutral differential equation. By using Riccati substitution and comparison technique, new oscillation conditions are obtained which insure that all solutions of the studied equation are oscillatory. Our results complement some known results for neutral differential equations. An illustrative example is included.

## 1. Introduction

In this paper, we study the oscillatory properties of solutions of the following fourth-order neutral differential equation
(1)$${\left(r\left(x\right){\left|w^{\prime \prime \prime }\left(x\right)\right|}^{p_{1}-2}w^{\prime \prime \prime }\left(x\right)\right)}^{\prime }+q\left(x\right){\left|y\left(g_{1}\left(x\right)\right)\right|}^{p_{2}-2}y\left(g_{1}\left(x\right)\right)=0,$$
where
(2)$$\begin{array}{c} w\left(x\right)=y\left(x\right)+\delta \left(x\right)y\left(g_{2}\left(x\right)\right) \end{array}$$
and subject to the following conditions:(*W*_1_)$p_{1},p_{2}>1\;$ are constants;(*W*_2_)$r,\delta \in C\left[x_{0},\infty \right),\;q\in C\left[x_{0},\infty \right),$$q\left(x\right)>0,\;r\left(x\right)>0,\;r^{\prime }\left(x\right)\geq 0,\;p_{2}\geq p_{1},$$0\leq \delta \left(x\right)<\delta_{0}<\infty ,$$q\left(x\right)$ is not identically zero for large *x*,(*W*_3_)$g_{2}\in C^{1}\left[x_{0},\infty \right),g_{1}\in C^{1}\left(\left[x_{0},\infty \right),R\right),$$g_{2}^{\prime }\left(x\right)>0,$$g_{1}^{\prime }\left(x\right)>0,$$g_{2}\left(x\right)\leq x,\lim_{x\to \infty }g_{2}\left(x\right)=\lim_{x\to \infty }g_{1}\left(x\right)=\infty$;(*W*_4_)$\int_{x_{0}}^{\infty }r^{-1/\left(p_{1}-1\right)}\left(s\right)ds=\infty .$

**Definition** **1.**
*A solution of (Equation 1) is said to be oscillatory if it has arbitrarily large zeros on $\left[x_{y},\infty \right)$. Otherwise, a solution that is not oscillatory is said to be nonoscillatory.*


**Definition** **2.**
*The Equation (Equation 1) is said to be oscillatory if every solution of it is oscillatory.*


**Definition** **3.**
*A differential equation is said to be neutral if the highest-order derivative of the unknown function appears both with and without delay.*


Neutral differential equations are used in numerous applications in technology and natural science. For instance, the problems of oscillatory behavior of neutral differential equations have a number of practical applications in the study of distributed networks containing lossless transmission lines which arise in high-speed computers where the lossless transmission lines are used to interconnect switching circuits, see [1,2,3,4]. In fact, half-linear differential equations arise in a variety of real world problems such as in the study of *p*-Laplace equations non-Newtonian fluid theory, the turbulent flow of a polytrophic gas in a porous medium, and so forth; see [5,6,7]. During the past few years there has been interest by many researchers to study the oscillatory behavior of this type of equation, see [8,9,10,11,12]. Furthermore, many researchers investigate regularity and existence properties of solutions to difference equations; see [13,14,15] and the references therein.

In [16], the authors studied oscillation conditions for equation
$${\left(r\left(t\right)\Phi \left(y^{\left(n-1\right)}\left(t\right)\right)\right)}^{\prime }+a\left(t\right)\Phi \left(y^{\left(n-1\right)}\left(t\right)\right)+q\left(t\right)\Phi \left(y\left(g\left(t\right)\right)\right)=0,$$
where $\Phi ={\left|s\right|}^{p-2}s$ and n is even. The authors used Riccati substitution together with integral averaging technique.

In [17], Bazighifan obtained oscillation conditions for solutions of (Equation 1) and used comparison method with second-order equations. Moreover, in [16,18,19], the authors considered the equation
$${\left(r\left(x\right){\left|w^{\prime \prime \prime }\left(x\right)\right|}^{p-2}w^{\prime \prime \prime }\left(x\right)\right)}^{\prime }+q\left(x\right){\left|y\left(\delta \left(x\right)\right)\right|}^{p-2}y\left(\delta \left(x\right)\right)=0,$$
where $w\left(x\right)=y\left(x\right)+\delta \left(x\right)y\left(g_{2}\left(x\right)\right)$ and obtained a condition under which every solution of this equation is oscillatory.

Bazighifan and Abdeljawad [20] give some results providing information on the asymptotic behavior of solutions of fourth-order advanced differential equations. This time, the authors used comparison method with first and second-order equations.

In this article, we establish oscillatory properties of solutions of (Equation 1). By using Riccati substitution and comparison technique, new oscillatory criteria for (Equation 1) are established. Our results complement some known results in literature. Furthermore, an illustrative example is provided.

## 2. Lemmas

The following lemmas will be used to establish our main results:

**Lemma** **1.**
*[21] Let β bea ratio of two odd numbers, D>0 and G are constants. Then*
$$Gy-Dy^{\left(\beta +1\right)/\beta }\leq \frac{\beta^{\beta }}{{\left(\beta +1\right)}^{\beta +1}}\frac{G^{\beta +1}}{D^{\beta }}.$$


**Lemma** **2.**
*[22] Let $y\in C^{n}\left(\left[x_{0},\infty \right),\left(0,\infty \right)\right).$ Assume that $y^{\left(n\right)}\left(x\right)$ is of fixed sign and not identically zero on $\left[x_{0},\infty \right)$ and that there exists a $x_{1}\geq x_{0}$ such that $y^{\left(n-1\right)}\left(x\right)y^{\left(n\right)}\left(x\right)\leq 0$ for all $x\geq x_{1}$. If $\lim_{x\to \infty }y\left(x\right)\neq 0,$ then for every $\mu \in \left(0,1\right)$ there exists $x_{\mu }\geq x_{1}$ such that*
$$y\left(x\right)\geq \frac{\mu }{\left(n-1\right)!}x^{n-1}\left|y^{\left(n-1\right)}\left(x\right)\right|\;for\;x\geq x_{\mu }.$$


**Lemma** **3.**
*[23] Let $y^{\left(i\right)}\left(x\right)>0,$i=0,1,..,n, and $y^{\left(n+1\right)}\left(x\right)<0\;$. Then*
$$\frac{y\left(x\right)}{x^{n}/n!}\geq \frac{y^{\prime }\left(x\right)}{x^{n-1}/\left(n-1\right)!}.$$


**Lemma** **4.**
*[3] Let c,v≥0 and β be a positive real number. Then*
$${\left(c+v\right)}^{\beta }\leq 2^{\beta -1}\left(c^{\beta }+v^{\beta }\right),\;for\;\beta \geq 1$$
*and*
$${\left(c+v\right)}^{\beta }\leq c^{\beta }+v^{\beta },\;for\;\beta \leq 1.$$


We consider the following notations:$$\begin{array}{ccc} \delta_{1}\left(x\right) & = & \frac{1}{\delta \left(g_{2}^{-1}\left(x\right)\right)}\left(1-\frac{{\left(g_{2}^{-1}\left(g_{2}^{-1}\left(x\right)\right)\right)}^{3}}{{\left(g_{2}^{-1}\left(x\right)\right)}^{3}\delta \left(g_{2}^{-1}\left(g_{2}^{-1}\left(x\right)\right)\right)}\right), \end{array}$$
$$\tilde{R}\left(x\right)={\left(\frac{\mu {\left(g_{2}^{-1}\left(\varphi \left(x\right)\right)\right)}^{3}}{6}\right)}^{\left(p_{2}-1\right)}q\left(x\right)\delta_{1}^{p_{2}-1}\left(\varphi \left(x\right)\right)r^{-\left(p_{2}-1\right)/\left(p_{1}-1\right)}\left(g_{2}^{-1}\left(\varphi \left(x\right)\right)\right),$$
$$R\left(x\right)=\int_{x}^{\infty }{\left(\frac{1}{r\left(\varrho \right)}\int_{\varrho }^{\infty }q\left(s\right){\left(\frac{g_{2}^{-1}\left(\sigma \left(s\right)\right)}{s}\right)}^{p_{2}-1}ds\right)}^{1/\left(p_{1}-1\right)}d\varrho$$
and
$$\begin{array}{c} \delta_{2}\left(x\right)=\frac{1}{\delta \left(g_{2}^{-1}\left(x\right)\right)}\left(1-\frac{\left(g_{2}^{-1}\left(g_{2}^{-1}\left(x\right)\right)\right)}{\left(g_{2}^{-1}\left(x\right)\right)\delta \left(g_{2}^{-1}\left(g_{2}^{-1}\left(x\right)\right)\right)}\right). \end{array}$$

The following lemma summarizes the situations that are discussed in the proofs of our results.

**Lemma** **5.**
*[24] Let y be an eventually positive solution of (Equation 1). Then there exist two possible cases:*
$$\begin{array}{ccc} & & \left.\left(S_{1}\right)\;\;w\left(x\right)>0,\;w^{\prime }\left(x\right)>0,\;w^{\prime \prime }\left(x\right)>0,\;w^{\prime \prime \prime }\left(x\right)>0,\;\;w^{\left(4\right)}\left(x\right)<0;\right.\\ & & \left.\left(S_{2}\right)\;w\left(x\right)>0,\;w^{\prime }\left(x\right)>0,\;w^{\prime \prime }\left(x\right)<0,\;w^{\prime \prime \prime }\left(x\right)>0,\;\;w^{\left(4\right)}\left(x\right)<0,\right. \end{array}$$

*for $\;x\geq x_{1},\;$ where $\;x_{1}\geq x_{0}\;$ is sufficiently large.*


## 3. Main Results

**Lemma** **6.**
*Let y be an eventually positive solution of (Equation 1). Then*
(3)$$y\left(x\right)\geq \frac{1}{\delta \left(g_{2}^{-1}\left(x\right)\right)}\left(w\left(g_{2}^{-1}\left(x\right)\right)-\frac{1}{\delta \left(g_{2}^{-1}\left(g_{2}^{-1}\left(x\right)\right)\right)}w\left(g_{2}^{-1}\left(g_{2}^{-1}\left(x\right)\right)\right)\right).$$


**Proof.** Let y be an eventually positive solution of (Equation 1). From the definition of *w*, we see that
$$\delta \left(x\right)y\left(g_{2}\left(x\right)\right)=w\left(x\right)-y\left(x\right)$$
and so
$$\delta \left(g_{2}^{-1}\left(x\right)\right)y\left(x\right)=w\left(g_{2}^{-1}\left(x\right)\right)-y\left(g_{2}^{-1}\left(x\right)\right).$$Repeating the same process, we obtain
$$y\left(x\right)=\frac{1}{\delta \left(g_{2}^{-1}\left(x\right)\right)}\left(w\left(g_{2}^{-1}\left(x\right)\right)-\left(\frac{w\left(g_{2}^{-1}\left(g_{2}^{-1}\left(x\right)\right)\right)}{\delta \left(g_{2}^{-1}\left(g_{2}^{-1}\left(x\right)\right)\right)}-\frac{y\left(g_{2}^{-1}\left(g_{2}^{-1}\left(x\right)\right)\right)}{\delta \left(g_{2}^{-1}\left(g_{2}^{-1}\left(x\right)\right)\right)}\right)\right),$$
which yields
$$y\left(x\right)\geq \frac{w\left(g_{2}^{-1}\left(x\right)\right)}{\delta \left(g_{2}^{-1}\left(x\right)\right)}-\frac{1}{\delta \left(g_{2}^{-1}\left(x\right)\right)}\frac{w\left(g_{2}^{-1}\left(g_{2}^{-1}\left(x\right)\right)\right)}{\delta \left(g_{2}^{-1}\left(g_{2}^{-1}\left(x\right)\right)\right)}.$$Thus, (Equation 3) holds. □

The first result of the paper is a theorem providing oscillation criterion for Equation (Equation 1). For this purpose, we employ the Riccati method.

**Theorem** **1.**
*Let $g_{1}\left(x\right)\leq g_{2}\left(x\right)$. Assume that there exist positive functions $\theta ,\theta_{1}\in C^{1}\left(\left[x_{0},\infty \right),R\right),$$\mu_{1}\in \left(0,1\right)$ and for every constants $M_{1},M_{2}>0$ such that*
(4)$$\begin{array}{cc} \; & \int_{x_{0}}^{\infty }\left(\eta_{1}\left(s\right)-\frac{2^{p_{1}-1}}{p_{1}^{p_{1}}}\frac{r\left(g_{2}^{-1}\left(g_{1}\left(s\right)\right)\right){\left(\theta^{\prime }\left(s\right)\right)}^{p_{1}}}{{\left(\mu_{1}\theta \left(s\right){\left(g_{2}^{-1}\left(g_{1}\left(s\right)\right)\right)}^{\prime }{\left(g_{1}\left(s\right)\right)}^{\prime }{\left(g_{2}^{-1}\left(g_{1}\left(s\right)\right)\right)}^{2}\right)}^{p_{1}-1}}\right)ds=\infty \end{array}$$
*and*
(5)$$\begin{array}{cc} \; & \int_{x_{0}}^{\infty }\left(\eta_{2}\left(s\right)-\frac{{\left(\theta_{1}^{\prime }\left(s\right)\right)}^{2}}{4\theta_{1}\left(s\right)}\right)ds=\infty , \end{array}$$
*where*
$$\begin{array}{ccc} \eta_{1}\left(x\right) & = & M_{1}^{p_{2}-p_{1}}\theta \left(x\right)q\left(x\right)\delta_{1}^{p_{2}-1}\left(g_{1}\left(x\right)\right) \end{array}$$
*and*
$$\begin{array}{ccc} \eta_{2}\left(x\right) & = & \delta_{2}^{\left(p_{2}-1\right)/\left(p_{1}-1\right)}\theta_{1}\left(x\right)M_{2}^{\left(p_{2}-p_{1}\right)/\left(p_{1}-1\right)}\int_{x}^{\infty }{\left(\frac{1}{r\left(\varrho \right)}\int_{\varrho }^{\infty }q\left(s\right){\left(\frac{g_{2}^{-1}\left(g_{1}\left(s\right)\right)}{s}\right)}^{p_{2}-1}ds\right)}^{1/\left(p_{1}-1\right)}d\varrho , \end{array}$$
*then (Equation 1) is oscillatory.*


**Proof.** Let *y* be a non-oscillatory solution of (Equation 1) on $\left[x_{0},\infty \right)$. Without loss of generality, we can assume that *y* is eventually positive. It follows from Lemma 5 that there exist two possible cases $\left(S_{1}\right)$ and $\left(S_{2}\right)$.Let $\left(S_{1}\right)$ hold. From Lemma 3, we obtain
$$\frac{w^{\prime }\left(x\right)}{w\left(x\right)}\leq \frac{3}{x}.$$Integrating from $g_{2}^{-1}\left(x\right)$ to *x*, we find
(6)$$\frac{w\left(g_{2}^{-1}\left(x\right)\right)}{w\left(x\right)}\geq \frac{{\left(g_{2}^{-1}\left(x\right)\right)}^{3}}{x^{3}}.$$This yields
(7)$${\left(g_{2}^{-1}\left(x\right)\right)}^{3}w\left(g_{2}^{-1}\left(g_{2}^{-1}\left(x\right)\right)\right)\leq {\left(g_{2}^{-1}\left(g_{2}^{-1}\left(x\right)\right)\right)}^{3}w\left(g_{2}^{-1}\left(x\right)\right).$$From (Equation 3) and (Equation 7), we get
(8)$$\begin{array}{ccc} y\left(x\right) & \geq & \frac{w\left(g_{2}^{-1}\left(x\right)\right)}{\delta \left(g_{2}^{-1}\left(x\right)\right)}\left(1-\frac{{\left(g_{2}^{-1}\left(g_{2}^{-1}\left(x\right)\right)\right)}^{3}}{{\left(g_{2}^{-1}\left(x\right)\right)}^{3}\delta \left(g_{2}^{-1}\left(g_{2}^{-1}\left(x\right)\right)\right)}\right)\\ & \geq & \delta_{1}\left(x\right)w\left(g_{2}^{-1}\left(x\right)\right). \end{array}$$From (Equation 1) and (Equation 8), we obtain
(9)$${\left(r\left(x\right){\left(w^{\prime \prime \prime }\left(x\right)\right)}^{p_{1}-1}\right)}^{\prime }+q\left(x\right)\delta_{1}^{p_{2}-1}\left(g_{1}\left(x\right)\right)w^{p_{2}-1}\left(g_{2}^{-1}\left(g_{1}\left(x\right)\right)\right)\leq 0.$$Define
(10)$$\psi_{1}\left(x\right):=\theta \left(x\right)\frac{r\left(x\right){\left(w^{\prime \prime \prime }\left(x\right)\right)}^{p_{1}-1}}{w^{p_{1}-1}\left(g_{2}^{-1}\left(g_{1}\left(x\right)\right)\right)}.$$From (Equation 9) and (Equation 10), we obtain
(11)$$\begin{array}{ccc} \psi_{1}^{\prime }\left(x\right) & \leq & \frac{\theta^{\prime }\left(x\right)}{\theta \left(x\right)}\psi_{1}\left(x\right)-\theta \left(x\right)w^{p_{2}-p_{1}}\left(g_{2}^{-1}\left(g_{1}\left(x\right)\right)\right)q\left(x\right)\delta_{1}^{p_{2}-1}\left(g_{1}\left(x\right)\right)\\ & & -\left(p_{1}-1\right)\theta \left(x\right)\frac{r\left(x\right){\left(w^{\prime \prime \prime }\left(x\right)\right)}^{p_{1}-1}{\left(g_{2}^{-1}\left(g_{1}\left(x\right)\right)\right)}^{\prime }{\left(g_{1}\left(x\right)\right)}^{\prime }w^{\prime }\left(g_{2}^{-1}\left(g_{1}\left(x\right)\right)\right)}{w^{p_{1}}\left(g_{2}^{-1}\left(g_{1}\left(x\right)\right)\right)}. \end{array}$$Recalling that $r\left(x\right){\left(w^{\prime \prime \prime }\left(x\right)\right)}^{p_{1}-1}$ is decreasing, we find
$$r\left(g_{2}^{-1}\left(g_{1}\left(x\right)\right)\right){\left(w^{\prime \prime \prime }\left(g_{2}^{-1}\left(g_{1}\left(x\right)\right)\right)\right)}^{p_{1}-1}\geq r\left(x\right){\left(w^{\prime \prime \prime }\left(x\right)\right)}^{p_{1}-1}.$$This yields
(12)$${\left(w^{\prime \prime \prime }\left(g_{2}^{-1}\left(g_{1}\left(x\right)\right)\right)\right)}^{p_{1}-1}\geq \frac{r\left(x\right)}{r\left(g_{2}^{-1}\left(g_{1}\left(x\right)\right)\right)}{\left(w^{\prime \prime \prime }\left(x\right)\right)}^{p_{1}-1}.$$It follows from Lemma 2 that
(13)$$w^{\prime }\left(g_{2}^{-1}\left(g_{1}\left(x\right)\right)\right)\geq \frac{\mu_{1}}{2}{\left(g_{2}^{-1}\left(g_{1}\left(x\right)\right)\right)}^{2}w^{\prime \prime \prime }\left(g_{2}^{-1}\left(g_{1}\left(x\right)\right)\right),$$
for all $\mu_{1}\in \left(0,1\right)$. Thus, by (Equation 11)–(Equation 13), we get
$$\begin{array}{ccc} \psi_{1}^{\prime }\left(x\right) & \leq & \frac{\theta^{\prime }\left(x\right)}{\theta \left(x\right)}\psi_{1}\left(x\right)-\theta \left(x\right)w^{p_{2}-p_{1}}\left(g_{2}^{-1}\left(g_{1}\left(x\right)\right)\right)q\left(x\right)\delta_{1}^{p_{2}-1}\left(g_{1}\left(x\right)\right)\\ & & -\left(p_{1}-1\right)\theta \left(x\right)\frac{\mu_{1}}{2}{\left(\frac{r\left(x\right)}{r\left(g_{2}^{-1}\left(g_{1}\left(x\right)\right)\right)}\right)}^{1/\left(p_{1}-1\right)}\frac{r\left(x\right){\left(w^{\prime \prime \prime }\left(x\right)\right)}^{p_{1}}{\left(g_{2}^{-1}\left(g_{1}\left(x\right)\right)\right)}^{\prime }{\left(g_{1}\left(x\right)\right)}^{\prime }{\left(g_{2}^{-1}\left(g_{1}\left(x\right)\right)\right)}^{2}}{w^{p_{1}}\left(g_{2}^{-1}\left(g_{1}\left(x\right)\right)\right)}. \end{array}$$Hence,
$$\begin{array}{ccc} \psi_{1}^{\prime }\left(x\right) & \leq & \frac{\theta^{\prime }\left(x\right)}{\theta \left(x\right)}\psi_{1}\left(x\right)-\theta \left(x\right)w^{p_{2}-p_{1}}\left(g_{2}^{-1}\left(g_{1}\left(x\right)\right)\right)q\left(x\right)\delta_{1}^{p_{2}-1}\left(g_{1}\left(x\right)\right)\\ & & -\left(p_{1}-1\right)\frac{\mu_{1}}{2}{\left(\frac{r\left(x\right)}{r\left(g_{2}^{-1}\left(g_{1}\left(x\right)\right)\right)}\right)}^{1/\left(p_{1}-1\right)}\frac{{\left(g_{2}^{-1}\left(g_{1}\left(x\right)\right)\right)}^{\prime }{\left(g_{1}\left(x\right)\right)}^{\prime }{\left(g_{2}^{-1}\left(g_{1}\left(x\right)\right)\right)}^{2}}{{\left(r\theta \right)}^{1/\left(p_{1}-1\right)}\left(x\right)}\psi_{1}^{\frac{p_{1}}{\left(p_{1}-1\right)}}\left(x\right). \end{array}$$Since $w^{\prime }\left(x\right)>0$, there exist $x_{2}\geq x_{1}$ and a constant M>0 such that
(14)$$w\left(x\right)>M,$$
for all $x\geq x_{2}$. Using Lemma 1, we put
$$G=\frac{\theta^{\prime }\left(x\right)}{\theta \left(x\right)},\;D=\left(p_{1}-1\right)\frac{\mu_{1}}{2}{\left(\frac{r\left(x\right)}{r\left(g_{2}^{-1}\left(g_{1}\left(x\right)\right)\right)}\right)}^{1/\left(p_{1}-1\right)}\frac{{\left(g_{2}^{-1}\left(g_{1}\left(x\right)\right)\right)}^{\prime }{\left(g_{1}\left(x\right)\right)}^{\prime }{\left(g_{2}^{-1}\left(g_{1}\left(x\right)\right)\right)}^{2}}{{\left(r\theta \right)}^{1/\left(p_{1}-1\right)}\left(x\right)}$$
and $y=\psi_{1}$, we get
$$\psi_{1}^{\prime }\left(x\right)\leq -\eta_{1}\left(x\right)+\frac{2^{p_{1}-1}}{p_{1}^{p_{1}}}\frac{r\left(g_{2}^{-1}\left(g_{1}\left(x\right)\right)\right){\left(\theta^{\prime }\left(x\right)\right)}^{p_{1}}}{{\left(\mu_{1}\theta \left(x\right){\left(g_{2}^{-1}\left(g_{1}\left(x\right)\right)\right)}^{\prime }{\left(g_{1}\left(x\right)\right)}^{\prime }{\left(g_{2}^{-1}\left(g_{1}\left(x\right)\right)\right)}^{2}\right)}^{p_{1}-1}}.$$This implies that
$$\int_{x_{1}}^{x}\left(\eta_{1}\left(s\right)-\frac{2^{p_{1}-1}}{p_{1}^{p_{1}}}\frac{r\left(g_{2}^{-1}\left(g_{1}\left(s\right)\right)\right){\left(\theta^{\prime }\left(s\right)\right)}^{p_{1}}}{{\left(\mu_{1}\theta \left(s\right){\left(g_{2}^{-1}\left(g_{1}\left(s\right)\right)\right)}^{\prime }{\left(g_{1}\left(s\right)\right)}^{\prime }{\left(g_{2}^{-1}\left(g_{1}\left(s\right)\right)\right)}^{2}\right)}^{p_{1}-1}}\right)ds\leq \psi_{1}\left(x_{1}\right),$$
which contradicts (Equation 4).Let $\left(S_{2}\right)$ hold. By using Lemma 3, we get
$$\frac{w^{\prime }\left(x\right)}{w\left(x\right)}\leq \frac{1}{x}.$$Integrating from $g_{2}^{-1}\left(x\right)$ to *x*, we find
(15)$$w\left(g_{2}^{-1}\left(g_{1}\left(x\right)\right)\right)\geq \frac{g_{2}^{-1}\left(g_{1}\left(x\right)\right)}{x}w\left(x\right).$$
which yields
(16)$$g_{2}^{-1}\left(x\right)w\left(g_{2}^{-1}\left(g_{2}^{-1}\left(x\right)\right)\right)\leq g_{2}^{-1}\left(g_{2}^{-1}\left(x\right)\right)w\left(g_{2}^{-1}\left(x\right)\right).$$From (Equation 3) and (Equation 16), we have
$$\begin{array}{ccc} y\left(x\right) & \geq & \frac{1}{\delta \left(g_{2}^{-1}\left(x\right)\right)}\left(1-\frac{\left(g_{2}^{-1}\left(g_{2}^{-1}\left(x\right)\right)\right)}{\left(g_{2}^{-1}\left(x\right)\right)\delta \left(g_{2}^{-1}\left(g_{2}^{-1}\left(x\right)\right)\right)}\right)w\left(g_{2}^{-1}\left(x\right)\right)\\ & = & \delta_{2}\left(x\right)w\left(g_{2}^{-1}\left(x\right)\right), \end{array}$$
which with (Equation 1) gives
(17)$${\left(r\left(x\right){\left(w^{\prime \prime \prime }\left(x\right)\right)}^{p_{1}-1}\right)}^{\prime }\leq -q\left(x\right)\delta_{2}^{p_{2}-1}\left(g_{1}\left(x\right)\right)w^{p_{2}-1}\left(g_{2}^{-1}\left(g_{1}\left(x\right)\right)\right).$$Integrating (Equation 17) from *x* to ϱ, we obtain
(18)$$r\left(\varrho \right){\left(w^{\prime \prime \prime }\left(\varrho \right)\right)}^{p_{1}-1}-r\left(x\right){\left(w^{\prime \prime \prime }\left(x\right)\right)}^{p_{1}-1}\leq -\int_{x}^{\varrho }\left(q\left(s\right)\delta_{2}^{p_{2}-1}\left(g_{1}\left(s\right)\right)w^{p_{2}-1}\left(g_{2}^{-1}\left(g_{1}\left(s\right)\right)\right)\right)ds.$$Letting ϱ→∞ in (Equation 18) and using (Equation 15), we obtain
$$r\left(x\right){\left(w^{\prime \prime \prime }\left(x\right)\right)}^{p_{1}-1}\geq \delta_{2}^{p_{2}-1}\left(g_{1}\left(x\right)\right)w^{p_{2}-1}\left(x\right)\int_{x}^{\infty }q\left(s\right){\left(\frac{g_{2}^{-1}\left(g_{1}\left(s\right)\right)}{s}\right)}^{p_{2}-1}ds.$$Integrating this inequality again from *x* to *∞*, we get
(19)$$w^{\prime \prime }\left(x\right)\leq -\delta_{2}^{\left(p_{2}-1\right)/\left(p_{1}-1\right)}w^{\left(p_{2}-1\right)/\left(p_{1}-1\right)}\left(x\right)\int_{x}^{\infty }{\left(\frac{1}{r\left(\varrho \right)}\int_{\varrho }^{\infty }q\left(s\right){\left(\frac{g_{2}^{-1}\left(g_{1}\left(s\right)\right)}{s}\right)}^{p_{2}-1}ds\right)}^{1/\left(p_{1}-1\right)}d\varrho ,$$Now, we define
$$\psi_{2}\left(x\right):=\theta_{1}\left(x\right)\frac{w^{\prime }\left(x\right)}{w\left(x\right)}.$$Then $\psi_{2}\left(x\right)>0$ for $x\geq x_{1}$. By differentiating $\psi_{2}$ and using (Equation 19), we find
$$\begin{array}{ccc} \psi_{2}^{\prime }\left(x\right) & = & \frac{\theta_{1}^{\prime }\left(x\right)}{\theta_{1}\left(x\right)}\psi_{2}\left(x\right)+\theta_{1}\left(x\right)\frac{w^{\prime \prime }\left(x\right)}{w\left(x\right)}-\theta_{1}\left(x\right){\left(\frac{w^{\prime }\left(x\right)}{w\left(x\right)}\right)}^{2}\\ & \leq & \frac{\theta_{1}^{\prime }\left(x\right)}{\theta_{1}\left(x\right)}\psi_{2}\left(x\right)-\frac{1}{\theta_{1}\left(x\right)}\psi_{2}^{2}\left(x\right)\\ & & -\delta_{2}^{\left(p_{2}-1\right)/\left(p_{1}-1\right)}\theta_{1}\left(x\right)w^{\left(p_{2}-p_{1}\right)/\left(p_{1}-1\right)}\left(x\right)\int_{x}^{\infty }{\left(\frac{1}{r\left(\varrho \right)}\int_{\varrho }^{\infty }q\left(s\right){\left(\frac{g_{2}^{-1}\left(g_{1}\left(s\right)\right)}{s}\right)}^{p_{2}-1}ds\right)}^{1/\left(p_{1}-1\right)}d\varrho . \end{array}$$Thus, we obtain
$$\psi_{2}^{\prime }\left(x\right)\leq -\eta_{2}\left(x\right)+\frac{\theta_{1}^{\prime }\left(x\right)}{\theta_{1}\left(x\right)}\psi_{2}\left(x\right)-\frac{1}{\theta_{1}\left(x\right)}\psi_{2}^{2}\left(x\right)$$Using Lemma 1, we put
$$G=\frac{\theta_{1}^{\prime }\left(x\right)}{\theta_{1}\left(x\right)},\;D=\frac{1}{\theta_{1}\left(x\right)}$$
and $y=\psi_{2}$, we find
$$\psi_{2}^{\prime }\left(x\right)\leq -\eta_{2}\left(x\right)+\frac{{\left(\theta_{1}^{\prime }\left(x\right)\right)}^{2}}{4\theta_{1}\left(x\right)}.$$Then, we get
$$\int_{x_{1}}^{x}\left(\eta_{2}\left(s\right)-\frac{{\left(\theta^{\prime }\left(s\right)\right)}^{2}}{4\theta \left(s\right)}\right)ds\leq \psi_{2}\left(x_{1}\right),$$
which contradicts (Equation 5). This completes the proof. □

The second result of the paper is a theorem providing oscillation criterion for Equation (Equation 1). For this purpose, we employ the comparison method with first-order differential equations.

**Theorem** **2.**
*Let*
(20)$$\frac{{\left(g_{2}^{-1}\left(g_{2}^{-1}\left(x\right)\right)\right)}^{3}}{{\left(g_{2}^{-1}\left(x\right)\right)}^{3}\delta \left(g_{2}^{-1}\left(g_{2}^{-1}\left(x\right)\right)\right)}\leq 1.$$

*Assume that there exist positive functions $\varphi ,\;\sigma \in \delta^{1}\left(\left[x_{0},\infty \right),R\right)$ satisfying*
(21)$$\varphi \left(x\right)\leq g_{1}\left(x\right),\;\varphi \left(x\right)<g_{2}\left(x\right),\;\sigma \left(x\right)\leq g_{1}\left(x\right),\;\sigma \left(x\right)<g_{2}\left(x\right),\;\sigma^{\prime }\left(x\right)\geq 0\;\mathrm{and}\;\lim_{x\to \infty }\varphi \left(x\right)=\lim_{x\to \infty }\sigma \left(x\right)=\infty .$$

*If there exists $\varepsilon_{1},\mu \in \left(0,1\right)$ such that the differential equations*
(22)$$u_{1}^{\prime }\left(x\right)+\tilde{R}\left(x\right)u_{1}^{\left(p_{2}-1\right)/\left(p_{1}-1\right)}\left(g_{2}^{-1}\left(\varphi \left(x,a\right)\right)\right)=0$$
*and*
(23)$$u_{2}^{\prime }\left(x\right)+\delta_{2}^{\left(p_{2}-1\right)/\left(p_{1}-1\right)}\varepsilon_{1}{\left(g_{2}^{-1}\left(\sigma \left(x\right)\right)\right)}^{\left(p_{2}-1\right)/\left(p_{1}-1\right)}R\left(x\right)u_{2}^{\left(p_{2}-1\right)/\left(p_{1}-1\right)}\left(g_{2}^{-1}\left(\sigma \left(x\right)\right)\right)=0$$

*are oscillatory, then (Equation 1) is oscillatory.*


**Proof.** Proceeding as in the proof of Theorem 1. Let Case $\left(S_{1}\right)$ hold. Since $\varphi \left(x\right)\leq g_{1}\left(x\right)$ and $w^{\prime }\left(x\right)>0$, we obtain
(24)$${\left(r\left(x\right){\left(w^{\prime \prime \prime }\left(x\right)\right)}^{p_{1}-1}\right)}^{\prime }\leq -q\left(x\right)\delta_{1}^{p_{2}-1}\left(\varphi \left(x\right)\right)w^{p_{2}-1}\left(g_{2}^{-1}\left(\varphi \left(x\right)\right)\right).$$Now, by using Lemma 2, we have
(25)$$w\left(x\right)\geq \frac{\mu }{6}x^{3}w^{\prime \prime \prime }\left(x\right),$$
for some $\mu \in \left(0,1\right)$. It follows from (Equation 24) and (Equation 25) that, for all $\mu \in \left(0,1\right),$
$${\left(r\left(x\right){\left(w^{\prime \prime \prime }\left(x\right)\right)}^{p_{1}-1}\right)}^{\prime }+{\left(\frac{\mu {\left(g_{2}^{-1}\left(\varphi \left(x\right)\right)\right)}^{3}}{6}\right)}^{p_{2}-1}q\left(x\right)\delta_{1}^{p_{2}-1}\left(\varphi \left(x\right)\right){\left(w^{\prime \prime \prime }\left(g_{2}^{-1}\left(\varphi \left(x\right)\right)\right)\right)}^{p_{2}-1}\leq 0.$$Thus, we choose
$$u_{1}\left(x\right)=r\left(x\right){\left(w^{\prime \prime \prime }\left(x\right)\right)}^{p_{1}-1}.$$So, we find that $\;u_{1}\;$ is a positive solution of the inequality
$$u_{1}^{\prime }\left(x\right)+\tilde{R}\left(x\right)u_{1}^{\left(p_{2}-1\right)/\left(p_{1}-1\right)}\left(g_{2}^{-1}\left(\varphi \left(x\right)\right)\right)\leq 0.$$Using (see [25], [Theorem 1]), we also see that (Equation 22) has a positive solution, a contradiction.Suppose that Case $\left(S_{2}\right)$ holds. From Theorem 1, we get that (Equation 19) holds. Since$\sigma \left(x\right)\leq g_{1}\left(x\right)$ and $w^{\prime }\left(x\right)>0$, we have that
(26)$$w^{\prime \prime }\left(x\right)\leq -\delta_{2}^{\left(p_{2}-1\right)/\left(p_{1}-1\right)}w^{\left(p_{2}-1\right)/\left(p_{1}-1\right)}\left(g_{2}^{-1}\left(\sigma \left(x\right)\right)\right)\int_{x}^{\infty }{\left(\frac{1}{r\left(\varrho \right)}\int_{\varrho }^{\infty }q\left(s\right){\left(\frac{g_{2}^{-1}\left(\sigma \left(s\right)\right)}{s}\right)}^{p_{2}-1}ds\right)}^{1/\left(p_{1}-1\right)}d\varrho .$$Using Lemma 3, we get that
(27)$$w\left(x\right)\geq \varepsilon_{1}xw^{\prime }\left(x\right).$$From (Equation 16) and (Equation 27), we obtain
$$w^{\prime \prime }\left(x\right)\leq -\delta_{2}^{\left(p_{2}-1\right)/\left(p_{1}-1\right)}\varepsilon_{1}{\left(w^{\prime }\left(g_{2}^{-1}\left(\sigma \left(x\right)\right)\right)\right)}^{\left(p_{2}-1\right)/\left(p_{1}-1\right)}{\left(g_{2}^{-1}\left(\sigma \left(x\right)\right)\right)}^{\left(p_{2}-1\right)/\left(p_{1}-1\right)}R\left(x\right).$$Now, we choose $u_{2}\left(x\right):=w^{\prime }\left(x\right)$, thus, we find that $u_{2}$ is a positive solution of
(28)$$u_{2}^{\prime }\left(x\right)+\delta_{2}^{\left(p_{2}-1\right)/\left(p_{1}-1\right)}\varepsilon_{1}{\left(g_{2}^{-1}\left(\sigma \left(x\right)\right)\right)}^{\left(p_{2}-1\right)/\left(p_{1}-1\right)}R\left(x\right)u_{2}^{\left(p_{2}-1\right)/\left(p_{1}-1\right)}\left(g_{2}^{-1}\left(\sigma \left(x\right)\right)\right)\leq 0.$$Using (see [25], [Theorem 1]), we also see that (Equation 23) has a positive solution, a contradiction. The proof is complete. □

**Example** **1.**
*Consider the equation*
(29)$$\left\{\begin{array}{cc} {\left(y\left(x\right)+16y\left(\frac{1}{2}x\right)\right)}^{\left(4\right)}+\frac{q_{0}}{x^{4}}y\left(\frac{1}{3}x\right)=0, & \\ t\geq 1,\;q_{0}>0. \end{array}\right.$$

*Let $\;r\left(x\right)=1,\;p_{1}=p_{2}=2,$$\delta \left(x\right)=16,$$g_{2}\left(x\right)=\frac{1}{2}x,\;g_{2}^{-1}\left(x\right)=2x,\;g_{1}\left(x\right)=\frac{1}{3}x$ and $q\left(t\right)=q_{0}/x^{4}$.*

*Thus, it is easy to see that*
$$\begin{array}{ccc} \delta_{1}\left(x\right) & = & \frac{1}{\delta \left(g_{2}^{-1}\left(x\right)\right)}\left(1-\frac{{\left(g_{2}^{-1}\left(g_{2}^{-1}\left(x\right)\right)\right)}^{3}}{{\left(g_{2}^{-1}\left(x\right)\right)}^{3}\delta \left(g_{2}^{-1}\left(g_{2}^{-1}\left(x\right)\right)\right)}\right)\\ & = & \frac{1}{16}\left(1-\frac{1}{2}\right)=\frac{1}{32},\\ \delta_{2}\left(x\right) & = & \frac{1}{\delta \left(g_{2}^{-1}\left(x\right)\right)}\left(1-\frac{\left(g_{2}^{-1}\left(g_{2}^{-1}\left(x\right)\right)\right)}{\left(g_{2}^{-1}\left(x\right)\right)\delta \left(g_{2}^{-1}\left(g_{2}^{-1}\left(x\right)\right)\right)}\right)\\ & = & \frac{1}{16}\left(1-\frac{1}{8}\right)=\frac{7}{128} \end{array}$$
*and*
$$\eta_{1}\left(x\right)=\frac{q_{0}}{32x},\eta_{2}\left(x\right)=\frac{7q_{0}}{1152x}.$$

*By applying conditions (Equation 4) and (Equation 5), we find*
$$\begin{array}{ccc} & & \int_{x_{0}}^{\infty }\left(\eta_{1}\left(s\right)-\frac{2^{p_{1}-1}}{p_{1}^{p_{1}}}\frac{r\left(g_{2}^{-1}\left(g_{1}\left(s\right)\right)\right){\left(\theta^{\prime }\left(s\right)\right)}^{p_{1}}}{{\left(\mu_{1}\theta \left(s\right){\left(g_{2}^{-1}\left(g_{1}\left(s\right)\right)\right)}^{\prime }{\left(g_{1}\left(s\right)\right)}^{\prime }{\left(g_{2}^{-1}\left(g_{1}\left(s\right)\right)\right)}^{2}\right)}^{p_{1}-1}}\right)ds\\ & = & \int_{x_{0}}^{\infty }\left(\frac{q_{0}}{32s}-\frac{729}{32s}\right)ds=\left(\frac{q_{0}}{32}-\frac{729}{32}\right)\int_{x_{0}}^{\infty }\frac{ds}{s}\\ & = & +\infty ,\;\;\;\;\;\;\;\;if\;\;\;q_{0}>729 \end{array}$$
*and for $\theta_{1}\left(x\right)=x,$ we get*
$$\begin{array}{ccc} \int_{x_{0}}^{\infty }\left(\eta_{2}\left(s\right)-\frac{{\left(\theta_{1}^{\prime }\left(s\right)\right)}^{2}}{4\theta_{1}\left(s\right)}\right)ds & = & \int_{x_{0}}^{\infty }\left(\frac{7q_{0}}{1152s}-\frac{1}{4s}\right)ds\\ & = & +\infty ,\;\;\;\;\;\;\;\;if\;\;\;q_{0}>41.14. \end{array}$$

*Hence, from Theorem 1, we conclude that (Equation 29) is oscillatory if $q_{0}>729$.*


## 4. Conclusions

In this work, by using the comparison and Riccati methods we establish a new oscillation conditions of (Equation 1). This new conditions complement some known results for neutral differential equations. Furthermore, in future work we will study the oscillatory behavior of this equation by comparison method with second-order differential equations. 

## Data Availability

Not applicable.

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
