# Peer review of "Nonlinear Neutral Delay Differential Equations of Fourth-Order: Oscillation of Solutions"

_entropy, 2021, doi:10.3390/e23020129_

Round 1
Reviewer 1 Report
I read this paper carefully and found it worthy and has merit. I would recommend major corrections before givie a decision:
1) I would ask the authors to enrich the intro section by adding a comparison paragraph. What is the relation between equation (1) and the results in [7] and [12] in terms of the equation and the used methods.
2) I need to understand why we need these classifications in Lemma 4. This should be justified.
3) Write the inequality in page 5 lines 36-37 as a separate lemma at intro section with reference.
4) I would like to see how authors use conditions H4 and H5 in the proof of Theorem2. This should be explained in details.
5) I attach a file on which authors may see my other comments that must be improved.
6) Please decrease self citations and add the following papers:
*https://doi.org/10.1186/s13662-019-2472-y
*On the oscillation of higher--order half--linear delay difference equations, Volume 6, No. 3 (Sep. 2012),PP:423-427. Applied Mathematics & Information Sciences

Author Response
Thank you very much for your comments and recommending our paper for
publication in the esteemed journal: Entropy after minor revisions.- In the introduction part, we added some comments in the introduction
on the relationship between the new results and those obtained in the
literature, we refer to previous established results and their importance.
We reworded some of the paragraphs.
- We need these classiÂ…cations in Lemma 5: To study the derivatives of the
equation.
- The inequality in page 5 lines 36-37 as a separate lemma with reference
has been written.
- Minor corrections: it has been modiÂ…ed.
- We have read again the paper and removed some english misprints.
- We added the references [R1]-[R2].
- We have checked once again computations and English language in all
results and example and made appropriate minor corrections to meet the
referee's Â’expectations.
Reviewer 2 Report
Sorry to say, the work lacks professionality and it contains serious flaws. For example, Theorem 1 does not obviously hold in this formulation. The hypothesis (H1) deals with the solution of the studied equation and cannot be in assumptions. Also the use of Lemmas and auxiliary statements is not done properly and without logical mistakes (e.g., in the second row of the proof of Thm 1 the use of Lemma 2 is rather peculiar, Lemma 4 does not seem to be correctly formulated,...).
Apart from logical faults in presenting the results and describing the proving process, there are other observations: some unclear formulations, language mistakes, missing or incorrect punctuation in the text and also in the references. It is not clear, why (H2) is assumed in Lemma 5. I find the work with "hypotheses" a bit unusual and not very helping the way they are presented.
Maybe, some parts of the proofs are well calculated, and some ideas can be used to study the equation, but the results must have different design.
Author Response
Thank you very much for your comments and recommending our paper for
publication in the esteemed journal: Entropy after major revisions.
- In the introduction part, we added some comments in the introduction
on the relationship between the new results and those obtained in the
literature, we refer to previous established results and their importance.
We reworded some of the paragraphs.
- We checked punctuation in whole the paper.
- Minor corrections: it has been modiÂ…ed.
- We have read again the paper and removed some english misprints.
- In the second row of the proof of Thm 1 the use of Lemma 2 when n = 3 (in case $ S_1 : \omega'' (x) > 0$ and $\omega''' (x) < 0$)
- Lemma 5 has been reformulated.
- About hypotheses:we totally agree with you, some hypotheses have been
removed.
- We have checked once again computations and English language in all
results and example and made appropriate minor corrections to meet the
referee's Â’expectations.
Reviewer 3 Report
The article presents results that appear in a sense of a movement on a much more general topic: Existence and oscillation of solutions to classes of differential equations with delay. The authors approach is based on some classical theories involving Riccati method and comparison techniques. The theoretical results seem to be correct and hence the manuscript can be published. Indeed, at the best of my knowledge, the results are new and complement the existing literature. I suggest the authors to carefully read the paper to correct a few of typos and language mistakes in whole the manuscript. For example:
1. Rephrase the second sentence of the abstract, writing "Riccati method and comparison techniques" instead of "different methods".
2. Change the key word "4th order" by "4th order differential equation".
3. Delete "In particular" at the beginning of Section 1.
4. I suggest to add a brief discussion about the content of references [17-24] and [3,4,11]. In particular, point out the the novelty in respect to the results established in [12].
5. Check punctuation in whole the paper (for example, at the end of page 2, line 3; at the end of equation (11) on page 4; at the end of page 4, line -1; at the end of equation (16) on page 5; and so on).
6. Delete ":" on page 3, line 3.
7. Check and correct the inequality on page 4, line 6 (that is, the line below inequality (9)).
8. Change "hypothesis" by "hypotheses" on page 6, line -1.
Author Response
Thank you very much for your comments and recommending our paper for
publication in the esteemed journal: Entropy after minor revisions.
- In the introduction part, we added some comments in the introduction
on the relationship between the new results and those obtained in the
literature, we refer to previous established results and their importance.
We reworded some of the paragraphs.
- Minor corrections: it has been modiÂ…ed.
- We have read again the paper and removed some english misprints.
- We checked punctuation in whole the paper.
- We have checked once again computations and English language in all
results and example and made appropriate minor corrections to meet the
refereesÂ’expectations.
We hope that revised version of the manuscript meets yours and the referee'sÂ’ expectations.
Round 2
Reviewer 1 Report
Please consider the attached report and try to correct all points. I need to see the revised version of round 2.

Author Response
Dear Referee,
thank you very much for your comments and recommending our paper for
publication in the esteemed journal Entropy after minor revisions.
Minor corrections: it has been modified.
- Indeed, we completely agree with you. There are some errors in the
proofs and assumptions, all errors have been corrected.
- We reformulated some Lemmas and Theorems and checked the proof.
- The style of the references have been unified.
- We have read again the paper and removed some english misprints.
- We reformulated some Lemmas and Theorems and checked the proof.
- We have checked once again computations and English language in all
results and example and made appropriate minor corrections to meet the
referees' expectations.
- We hope that revised version of the manuscript meets yours and the referees'
expectations
Best regards and thanks again
Reviewer 2 Report
In the paper, there are still too many mistakes and unjustified places.
Major issues
Main results are not properly formulated, the proofs contain places, where deeper description or correction is needed. The example does not deal with the relevant type of equation (4th order).
More concretely:
Theorem 1 - the assumption of positivity of a solution y cannot be a part of the statement. Instead of "every M1,M2", there must be something else (e.g. "constants M1,M2...")
Proof of Theorem 1. The use of Lemma 3 in the beginning needs the condition w''''<=0. This is not validated. Moreover, the implication "hence x^{-3}w(x) is nonincreasing" is not clear. How does it follow from the previou inequality?
What is "n" in the inequality (7)?
The sentence after (8) does not make sense, the relation below only repeats (8), there is nothing new.
After (10), condition "g2(x) \geq g1(x)" is probably needed. From where does it follow?
The use of Lemma 2 - again the condition w''''<=0 is needed.
The sentence before (13): remove "a" and what is x1? It has not been used yet.
Below (13) I think also condition g1'>0 is needed but not supposed (for D to be positive).
8th row from below p. 6: what is "alpha"?
6th row from below: lower limit of the integral should be x0?
by (14) comment on epsilon1
(16): on the right side, the power of w is missing (p2-1)
I think, the shift from (18) to the next relation needs some conditions for putting powers of w(x) and delta2(x) in front of the integral. Consider commenting and verifying the conditions.
After (19), what is "mu2"?
5th row from below p. 7: power of w should be different
End of the 7th page, clarification of the implication commented as "and so" is needed. From where does it follow?
Theorem 2 - formulation without "hypotheses" would be better (as has been done in Theorem 1).
Proof of Theorem 2. - 2nd row - what is "z"?
Use of Lemma 2 needs w'''' <=0.
p.9, 2nd row - in the middle, there is an extra "z", by the end, w should not be primed?
Example 1 - what is "n"?
Formal issues
Misprints: equation (1) - y without primes
Language: "conditions complement" 3 times
Proof of Lemma 6: Let y "be"
Before both theorems "... an oscillation criterion for ..."
Formulation of hypotheses, formulation of the last sentence in the conclusion. In the Conclusion also "...establish new oscillation conditions..."
The style of the references is not unified.
Advice: If the paper is a work of Ph.D. students, I recommend to let it be read by the supervisor first.
Author Response
Dear Referee,
thank you very much for your comments and recommending our paper for
publication in the esteemed journal Entropy after minor revisions.
Minor corrections 1-33: it has been modified.
We have read again the paper and removed some english misprints.
We reformulated some Lemmas and Theorems and checked the proof.
We added the references https://doi.org/10.1186/s13662-020-03156-0.
We have checked once again computations and English language in all
results and example and made appropriate minor corrections to meet the
referees' expectations.We hope that revised version of the manuscript meets yours and the referees' expectations.
Thanks and Best Regards.
Round 3
Reviewer 1 Report
I can say now that the paper is accepted and can be published as it is.
Author Response
The authors are grateful for the fruitful revision that improves the paper.
Best regards
Reviewer 2 Report
Dear authors, in your last reply and in the new version of the paper, only a part of my suggestions and question has been considered, still there are plenty of unclear places. Please, see again the comments in the attached file. If I'm mistaken (which might happen), explain me those parts in your reply, if not, correct those places. Please, see all of them, handwritten in green.
If the paper is reconsidered again, I would like to see the new version.

Author Response
Dear Editors,
we would like to thank oll of you for useful comments that helped us to improve the original manuscript; this has also been acknowledged in the revised version of the paper.
Now, we are sending the revised article for your reconsideration to publish
in Entropy. Please, see our point to point responses to the comments
below, and the corresponding revisions in the body of manuscript. We look
forward to hearing from you soon for a favorable decision.
Thank you very much for your comments and recommending our paper for
publication in the esteemed journal Entropy after minor revisions.
- All errors have been corrected in the proofs and assumptions.
- We have read again the paper and removed some english misprints.
- We added some important definitions in the introduction.
- We explained some mathematical relationships and inequalities.
- About Theorem 1: We reformulated Theorem 1 and checked the proof.
Also, we used Lemma 3 in cases $S_{1}$ and $S_{2}$, then we Integrating from
$g_{2}^{-1}(x)$ to $x$, and this is clear and easy.
- About Lemma 5: two possible cases are correct, see paper: Bazighifan
O, Ruggieri M, Scapellato A. An Improved Criterion for the Oscillation of
Fourth-Order Differential Equations. Mathematics. 2020; 8(4):610.
-The shift from (18) to the next relation is correct.
-We added condition $g_{1}^{^{\prime }}(x)>0.$
-The results are correct, for some comments you can see the articles:
Bazighifan O, Ruggieri M, Scapellato A. An Improved Criterion for the
Oscillation of Fourth-Order Differential Equations. Mathematics. 2020;
8(4):610.
Zhang, C.; Agarwal, R.P.; Bohner, M.; Li, T. New results for oscillatory
behavior of even-order half-linear delay differential equations. Appl. Math.
Lett. 2013, 26, 179--183.
Zhang, C.; Li, T.; Suna, B.; Thandapani, E. On the oscillation of
higher-order half-linear delay differential equations. Appl. Math. Lett.
2011, 24, 1618--1621.
Bazighifan, O.; Moaaz, O.; El-Nabulsi, R.A.; Muhib, A. Some New Oscillation
Results for Fourth-Order Neutral Differential Equations with Delay Argument.
Symmetry 2020, 12, 1248.
We have checked once again computations and English language in all
results and example and made appropriate minor corrections to meet the
referees' expectations.
Thanks and Best Regards.
Sincerely,
Omar and Maria Alessandra